# Effects of Adrenomedullin on Atrial Electrophysiology and Pulmonary Vein Arrhythmogenesis

**DOI:** 10.3390/ijms232214064

**Published:** 2022-11-15

**Authors:** Chye-Gen Chin, Ahmed Moustafa Elimam, Fong-Jhih Lin, Yao-Chang Chen, Yung-Kuo Lin, Yen-Yu Lu, Satoshi Higa, Shih-Ann Chen, Ming-Hsiung Hsieh, Yi-Jen Chen

**Affiliations:** 1Graduate Institute of Clinical Medicine, College of Medicine, Taipei Medical University, Taipei 11042, Taiwan; 2Division of Cardiovascular Medicine, Department of Internal Medicine, Wan Fang Hospital, Taipei Medical University, 111 Hsin-Lung Road, Sec. 3, Taipei 11696, Taiwan; 3Division of Cardiovascular Medicine, Department of Internal Medicine, Mansoura International Hospital, Mansoura 35511, Egypt; 4Department of Biomedical Engineering, National Defense Medical Center, Taipei 11490, Taiwan; 5Division of Cardiology, Department of Internal Medicine, School of Medicine, College of Medicine, Taipei Medical University, Taipei 11042, Taiwan; 6Division of Cardiology, Department of Internal Medicine, Sijhih Cathay General Hospital, New Taipei City 22174, Taiwan; 7Cardiac Electrophysiology and Pacing Laboratory, Division of Cardiovascular Medicine, Makiminato Central Hospital, Okinawa 901-2131, Japan; 8Department of Post-Baccalaureate Medicine, College of Medicine, National Chung Hsing University, Taichung 40227, Taiwan; 9Cardiovascular Center, Taichung Veterans General Hospital, Taichung 40705, Taiwan

**Keywords:** atrial fibrillation, adrenomedullin, pulmonary vein, calcium, heart failure

## Abstract

Adrenomedullin, a peptide with vasodilatory, natriuretic, and diuretic effects, may be a novel agent for treating heart failure. Heart failure is associated with an increased risk of atrial fibrillation (AF), but the effects of adrenomedullin on atrial arrhythmogenesis remain unclear. This study investigated whether adrenomedullin modulates the electrophysiology of the atria (AF substrate) or pulmonary vein (PV; AF trigger) arrhythmogenesis. Conventional microelectrode or whole-cell patch clamps were used to study the effects of adrenomedullin (10, 30, and 100 pg/mL) on the electrical activity, mechanical response, and ionic currents of isolated rabbit PV and sinoatrial node tissue preparations and single PV cardiomyocytes. At 30 and 100 pg/mL, adrenomedullin significantly reduced the spontaneous beating rate of the PVs from 2.0 ± 0.4 to 1.3 ± 0.5 and 1.1 ± 0.5 Hz (reductions of 32.9% ± 7.1% and 44.9 ± 8.4%), respectively, and reduced PV diastolic tension by 12.8% ± 4.1% and 14.5% ± 4.1%, respectively. By contrast, adrenomedullin did not affect sinoatrial node beating. In the presence of L-NAME (a nitric oxide synthesis inhibitor, 100 μM), adrenomedullin (30 pg/mL) did not affect the spontaneous beating rate or diastolic tension of the PVs. In the single-cell experiments, adrenomedullin (30 pg/mL) significantly reduced the L-type calcium current (I_Ca-L_) and reverse-mode current of the sodium–calcium exchanger (NCX). Adrenomedullin reduces spontaneous PV activity and PV diastolic tension by reducing I_Ca-L_ and NCX current and thus may be useful for treating atrial tachyarrhythmia.

## 1. Introduction

Adrenomedullin is a 52-amino acid peptide of the calcitonin gene-related peptide family. Adrenomedullin has several cardiovascular effects (vasodilatation, natriuresis, diuresis, promotion of vascular integrity, and prevention of vascular leakage) [1,2,3,4,5] and non-cardiovascular effects (anti-inflammatory activity, mucosal epithelial repair, and maintenance of intestinal barrier function) [6,7]. Adrenomedullin and its receptors are widely distributed in various tissues and highly expressed in blood vessels. Because of its vasodilatory effect and ubiquitous expression in the cardiovascular system, it was considered a target for treating cardiovascular diseases, including heart failure (HF). Plasma adrenomedullin level was correlated with the severity of HF and may prevent HF progression. Several reports have shown that plasma adrenomedullin levels increase proportionally with HF severity [8,9]. Relationships between the plasma levels of adrenomedullin and other neurohumoral factors, including atrial natriuretic peptide, brain (or B-type) natriuretic peptide, and norepinephrine [9], have been reported. A phase 1 clinical trial indicated that adrenomedullin at 15 ng/kg/min may be effective for treating HF by reducing pulmonary edema through the stabilization of endothelial barrier function without notable hemodynamic or humoral effects [10]. A crucial function of adrenomedullin is its dilatation of both resistance and capacitance vessels. This vasodilatory effect is mediated by the binding of adrenomedullin with its receptors on vascular endothelial cells and vascular smooth muscle cells [11], which increases intracellular cyclic adenosine monophosphate [12] and activates protein kinase A (PKA) [11].

HF is a major risk factor for atrial fibrillation (AF), and AF is the most common sustained arrhythmia in patients with HF, with an average prevalence of 25% [13]. HF increases the arrhythmogenicity of the pulmonary veins (PVs), which are the largest sources of ectopic beats initiating paroxysmal AF and the foci of ectopic atrial tachycardia [14]. Increases in atrial pressure accelerate the firing rates of the PVs, leading to AF [15]. Adrenomedullin was demonstrated to reduce the I_Ca-L_ in the myocytes of rats with septic shock through the PKA signaling pathway [16]. Because PKA signaling plays a critical role in AF pathogenesis, adrenomedullin may modulate atrial arrhythmogenesis. Our previous study revealed that the activation of PKA signaling may regulate isoproterenol-induced PV arrhythmogenesis. However, the effects of adrenomedullin on atrial arrhythmogenesis remain unclear. Therefore, adrenomedullin may reduce the risk of AF through effects on HF or PV arrhythmogenesis. This study investigated whether adrenomedullin modulates the electrophysiology of the atrium (AF substrate) or PV arrhythmogenesis.

## 2. Results

### 2.1. Effects of Adrenomedullin on PV and SAN (Sinoatrial Node) Electrical Activity

As depicted in Figure 1, the administration of 30 and 100 pg/mL adrenomedullin reduced the PV spontaneous beating rate in a dose-dependent manner; specifically, 30 and 100 pg/mL resulted in reductions of 32.9% ± 7.1% and 44.9% ± 8.4%, respectively. As presented in Figure 1B, the administration of adrenomedullin (10, 30, and 100 pg/mL) did not significantly affect the SAN spontaneous beating rate. In addition, adrenomedullin (10, 30, and 100 pg/mL) significantly reduced the rates of spontaneous, early, and late diastolic depolarization in PV tissue (Figure 2A). However, adrenomedullin (10–100 pg/mL) did not affect the maximum diastolic potential or maximum upstroke velocity in PV tissue. Additionally, adrenomedullin at 30 and 100 pg/mL reduced PV tone by 12.8% ± 4.1% and 14.5% ± 4.1%, respectively (Figure 2B). However, in the presence of L-NAME (100 μmol/L), 30 pg/mL adrenomedullin did not reduce the spontaneous beating rate or diastolic tension of the PVs (*n* = 5; Figure 3).

### 2.2. Effects of Adrenomedullin on Left Atrium (LA) and Right Atrium (RA) Electrophysical Characteristics and Contractility

Adrenomedullin at 30 and 100 pg/mL (but not 10 pg/mL) reduced LA action potentials duration (APD)_50_ (Figure 4), and adrenomedullin at 100 pg/mL reduced LA APD_20_. However, adrenomedullin (10, 30, or 100 pg/mL) did not alter LA resting membrane potential (RMP) or APD_90_. Adrenomedullin at 100 pg/mL reduced RA APD_20_ and APD_50_ (Figure 5), but adrenomedullin (10, 30 pg/mL) did not alter the action potentials amplitude (APA), RMP, or APD_90_ of the RA. Finally, adrenomedullin did not reduce the contractility of the LA or RA (Figure 4 and Figure 5).

### 2.3. Effects of Adrenomedullin on I_Ca-L_ and NCX in PV Cardiomyocytes

In order to clarify whether regulation of calcium influx may contribute to the effects of adrenomedullin on PV electrical activity, we investigated the effects of adrenomedullin on I_Ca-L_ and sodium–calcium exchanger (NCX) in PV cardiomyocytes. We found that adrenomedullin (30 pg/mL) significantly reduced the I_Ca-L_ in PV cardiomyocytes, resulting in a 37.6% ± 4.7% decrease in peak I_Ca-L_ (elicited from −40 to +10 mV). However, adrenomedullin increased the forward- and reverse-mode NCX currents (Figure 6).

## 3. Discussion

Adrenomedullin is a biomarker for coronary artery disease and HF [8,17], but little is known about the peptide in AF. In normal human cardiac tissue, the positive inotropic effects of adrenomedullin occur mainly in the atria; these effects are reflected by increased contractile force. During atrial stretch, the adrenomedullin signaling cascade is downregulated, increasing susceptibility to AF [18]. However, adrenomedullin appears unsuitable for predicting recurrent [19] or incident [20] AF events.

In this study, we observed that 30 and 100 pg/mL adrenomedullin significantly reduced spontaneous PV activity. Previous studies have shown that the average plasma concentrations of adrenomedullin were around 7.2 pg/mL in healthy individuals [21] and around 33.8 pg/mL in patients with HF [8], respectively. The treatment of adrenomedullin may increase the concentrations of serum adrenomedullin to three times higher than that at baseline [22]. Thus, the concentrations of adrenomedullin used in this study were considered clinically relevant. Adrenomedullin exerts an endothelium-dependent vasodilatory effect through the nitric oxide (NO)–cyclic guanosine monophosphate pathway in rat aortae, the renal and hindquarter vascular beds of rats, and kidneys of dogs [23,24]. We observed decreases in vascular tone after the administration of 30 and 100 pg/mL adrenomedullin. Stretch-induced mechanoelectrical feedback regulates PV arrhythmogenesis [25]; thus, adrenomedullin may reduce PV arrhythmogenesis, at least in part through its inhibitory effects on vascular diastolic tension (mechanoelectrical feedback). Moreover, the NO synthesis inhibitor L-NAME suppressed antiarrhythmogenesis in the adrenomedullin-treated PV tissue preparations, suggesting that adrenomedullin may activate NO signaling.

Adrenomedullin can induce an initial increase in cell shortening and Ca^2+^ transients after a short (30 min) incubation followed by marked decreases in cell shortening and Ca^2+^ transients [26,27]. In addition, adrenomedullin reduced Ca^2+^ and I_Ca-L_ in rabbit ventricular myocytes [28]. In this study, we observed that adrenomedullin directly reduced I_Ca-L_ in PV cardiomyocytes, and such reduction may contribute to its effect on spontaneous PV activity. We also observed that adrenomedullin significantly shortened the APD_20_ and APD_50_ but not the APD_90_ of the LA and RA. These results suggest that the shortening of the APD_20_ and APD_50_ of the atria is mainly attributable to the effects of adrenomedullin on I_Ca-L_. Our findings suggest that adrenomedullin reduces AF risk through its inhibition of PV, LA, and RA arrhythmogenesis.

We observed that 100 pg/mL adrenomedullin only mildly reduced spontaneous SAN activity. These mild effects on the SAN may prevent bradycardia during adrenomedullin administration, which is a critical factor in atrial arrhythmogenesis, because the loss of SAN overdrive modulation of subsidiary pacemaker cells may facilitate the generation of ectopic arrhythmias. However, the mechanisms underlying the different responses of the PVs and SAN to adrenomedullin are unclear and may be caused by different characteristics underlying spontaneous SAN and PV activity because pacemaker current plays a major role in SAN activity but contributes little to PV activity [29]. Moreover, adrenomedullin-induced dilation of the PVs may exert antiarrhythmic potential by reducing spontaneous PV activity through mechanoelectrical feedback.

## 4. Materials and Methods

### 4.1. Electropharmacological Studies of PV and Sinoatrial Node Tissues

All experiments were conducted in accordance with the *Guide for the Care and Use of Laboratory Animals* published by the United States National Research Council Institute for Laboratory Animal Research (animal permission number: LAC-2021-0381). Male New Zealand rabbits (average age of 6 months and weighing 2.5–3.0 kg) were anesthetized with inhalational isoflurane (2.0–2.5%) from a precision vaporizer for 10 min. The adequacy of anesthesia was confirmed by a lack of corneal reflexes or motor response to the painful stimulus of a scalpel tip. The SAN was dissected from the junction of the superior vena cava and RA. PV tissues were dissected from the LA at the LA–PV junction and from the lungs at the end of the PV myocardial sleeve. The LA–PV-junction end of the tissue preparations was attached with needles at the bottom of a tissue bath, and the other (distal PV) end was connected to a Grass FT03C force transducer (Grass Instruments, Beverly, MA, USA) by silk thread. As described previously [30], the LA and RA tissue preparations (1 × 1.5 cm^2^) were separated from the LA and RA appendages, respectively. The LA, RA, PV, and SAN tissues were perfused at a constant rate (3 mL/min) with Tyrode solution containing NaCl (137 mmol/L), KCl (4 mmol/L), NaHCO_3_ (15 mmol/L), NaH_2_PO_4_ (0.5 mmol/L), MgCl_2_ (0.5 mmol/L), CaCl_2_ (2.7 mmol/L), and glucose (11 mmol/L) and then saturated with a gas mixture containing 97% O_2_ and 3% CO_2_. The LA, RA, PV, and SAN tissues were treated before and after the acute administration of various concentrations (10, 30, and 100 pg/mL) of adrenomedullin with or without 100 μM L-NAME (an NO synthesis inhibitor) for analysis of the electrophysiological effects of adrenomedullin.

The transmembrane action potentials of the LA, RA, PV, and SAN tissues were recorded using machine-pulled glass capillary microelectrodes filled with 3 mol/L KCl, connected to a World Precision Instruments Duo 773 Electrometer (Sarasota, FL, USA) under tension of 150 mg. The temperature was maintained at 37 °C, and the tissue preparations were allowed to equilibrate for 1 h before the electrophysiological and pharmacological evaluations. Electrical and mechanical events were displayed on a Gould 4072 Oscilloscope (Gould Electronics, Cleveland, OH, USA) and simultaneously recorded with a Gould TA11 Recorder as described previously [31]. The APA was calculated as the difference between the RMP and peak of AP depolarization. The (APD) at 90%, 50%, and 20% repolarization was measured and designated as APD_90_, APD_50_, and APD_20_, respectively.

### 4.2. Isolation of Single PV Cardiomyocytes

Single PV cardiomyocytes were isolated from the rabbits (2.0–3.0 kg) as described previously [31]. A whole-cell patch clamp was used to record ionic currents in the isolated PV cardiomyocytes with an Axopatch 1D amplifier (Axon Instruments, Foster City, CA, USA) at 35 ± 1 °C. The ionic currents were recorded approximately 3–5 min after rupture or perforation to avoid any decay in ion channel activity. At the beginning of each experiment, a small hyperpolarizing step from a holding potential of −50 mV to a test potential of −55 mV for 80 ms was performed. The area under the capacitive current curve was divided by the applied voltage step to obtain the total cell capacitance.

I_Ca-L_ was measured for 300 ms at a frequency of 0.1 Hz as the inward current during depolarization from a holding potential of −50 mV with test potentials increasing from −40 to +60 mV with 10 mV steps. Measurement was conducted using a perforated patch clamp with an external solution that contained tetraethylammonium chloride (137 mmol/L), CsCl (5.4 mmol/L), MgCl_2_ (0.5 mmol/L), CaCl_2_ (1.8 mmol/L), HEPES (10 mmol/L), and glucose (10 mmol/L), which was adjusted to pH 7.4 with CsOH. Micropipettes were filled with a solution containing CsCl (130 mmol/L), MgCl_2_ (1 mmol/L), MgATP (5 mmol/L), HEPES (10 mmol/L), Na-guanosine triphosphate (0.1 mmol/L), and Na_2_ phosphocreatine (5 mmol/L), which was adjusted to pH 7.2 with CsOH.

The NCX current was elicited through the administration of test pulses of between −100 and +100 mV from a holding potential of −40 mV for 300 ms at a frequency of 0.1 Hz. The amplitudes of the NCX current were measured using 10 mM nickel-sensitive currents. The external solution consisted of NaCl (140 mM), CaCl_2_ (2 mM), MgCl_2_ (1 mM), HEPES (5 mM), and glucose (10 mM) at pH 7.4 as well as strophanthidin (10 μM), nitrendipine (10 μM), and niflumic acid (100 μM).

### 4.3. Statistical Analysis

All continuous variables are expressed as means ± standard deviations. A paired *t* test or one-way repeated-measures analysis of variance with a Bonferroni post hoc test was employed to compare variable values before and after the administration of adrenomedullin and L-NAME. A *p* value < 0.05 was considered statistically significant.

## 5. Conclusions

Adrenomedullin may activate NO signaling and exert antiarrhythmic effects by reducing spontaneous PV activity through mechanoelectrical feedback, reducing I_Ca-L_ and NCX current. Adrenomedullin administration represents a potential novel strategy for management of atrial arrhythmogenesis.

## Figures and Tables

**Figure 1 ijms-23-14064-f001:**
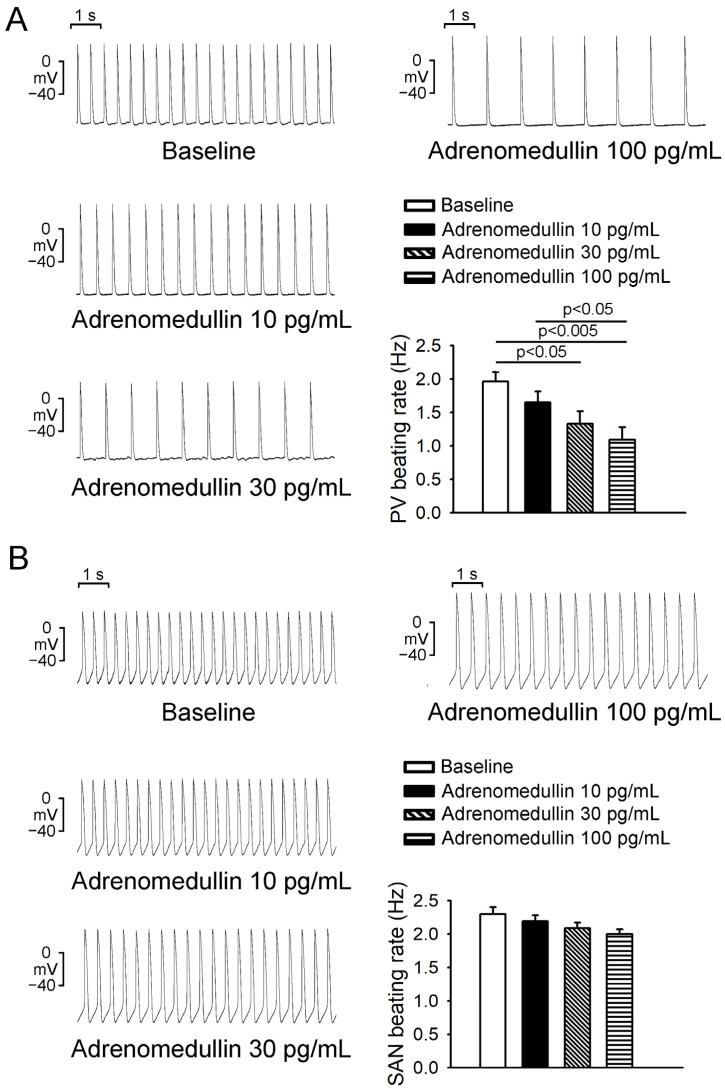
Effects of adrenomedullin on pulmonary vein (PV) and sinoatrial node electrical activity. (**A**) Adrenomedullin at 30 and 100 pg/mL reduced PV (*n* = 8) beating rates. (**B**) Adrenomedullin (10–100 pg/mL) did not reduce SN (*n* = 8) spontaneous beating rates.

**Figure 2 ijms-23-14064-f002:**
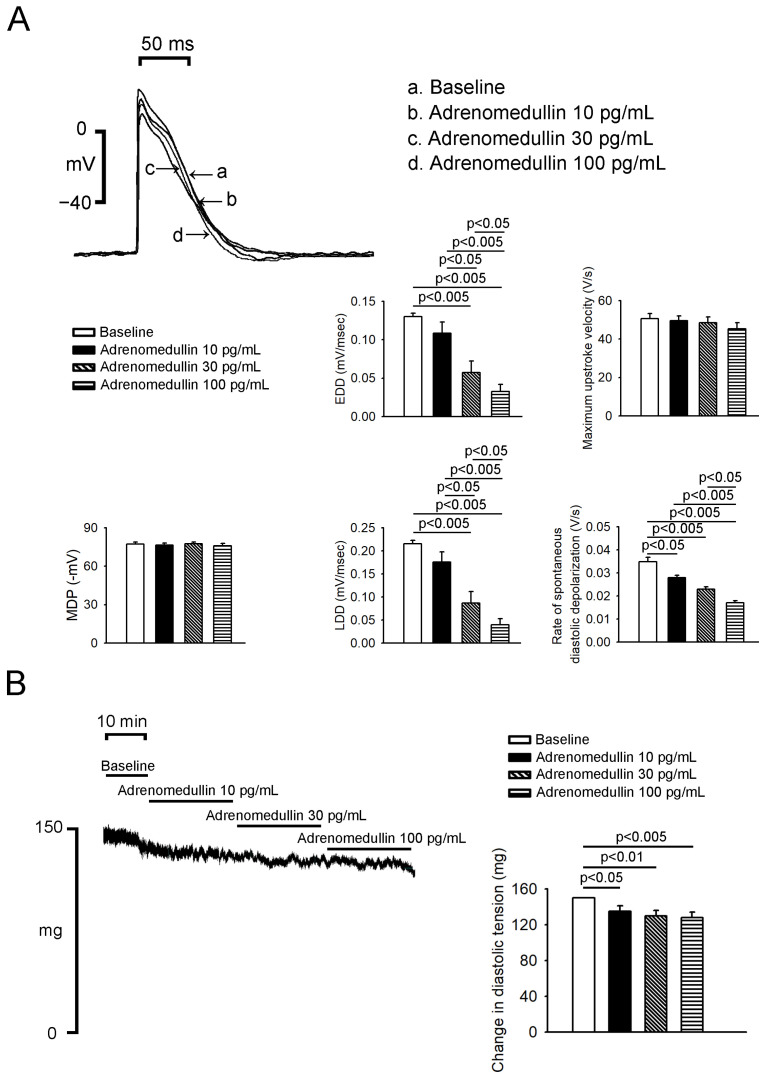
Effects of adrenomedullin on PV action potential (AP) morphology and vascular tone. (**A**) The superimposed traces illustrate the effects of adrenomedullin on the PV APs. Adrenomedullin (10–100 pg/mL) reduced the rates of spontaneous, early, and late diastolic depolarization. However, adrenomedullin (10–100 pg/mL) did not affect the maximum diastolic potential (MDP) or maximum upstroke velocity in PV tissue (*n* = 8). (**B**) Adrenomedullin reduced PV (*n* = 8) diastolic tension.

**Figure 3 ijms-23-14064-f003:**
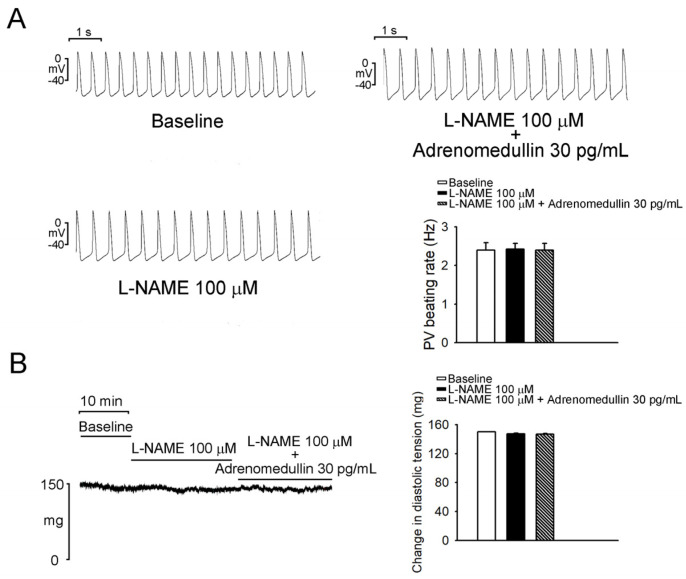
Interactions of L-NAME and adrenomedullin on PV electrical activity. (**A**) Adrenomedullin at 30 pg/mL did not reduce PV beating rates in the presence of 100 μmol/L L-NAME. (**B**) Adrenomedullin at 30 pg/mL did not reduce PV diastolic tension in the presence of 100 μmol/L L-NAME.

**Figure 4 ijms-23-14064-f004:**
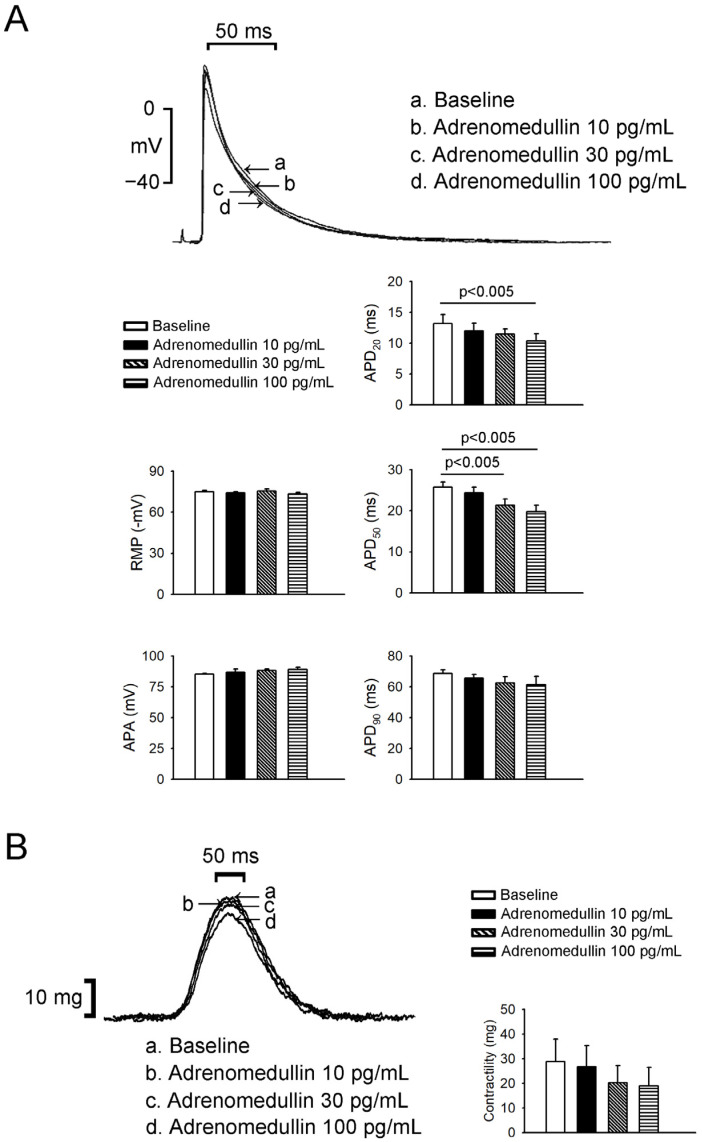
Effects of adrenomedullin on left atrium (LA) AP morphology and contractility. (**A**) The superimposed traces illustrate the effects of adrenomedullin on LA APs. Adrenomedullin (10–100 pg/mL) did not change APA, resting membrane potential, or APD90 in LA tissue (*n* = 7). Adrenomedullin at 30 and 100 pg/mL (but not 10 pg/mL) reduced LA AP duration at 20% and 50% repolarization. (**B**) Adrenomedullin did not reduce LA contractility.

**Figure 5 ijms-23-14064-f005:**
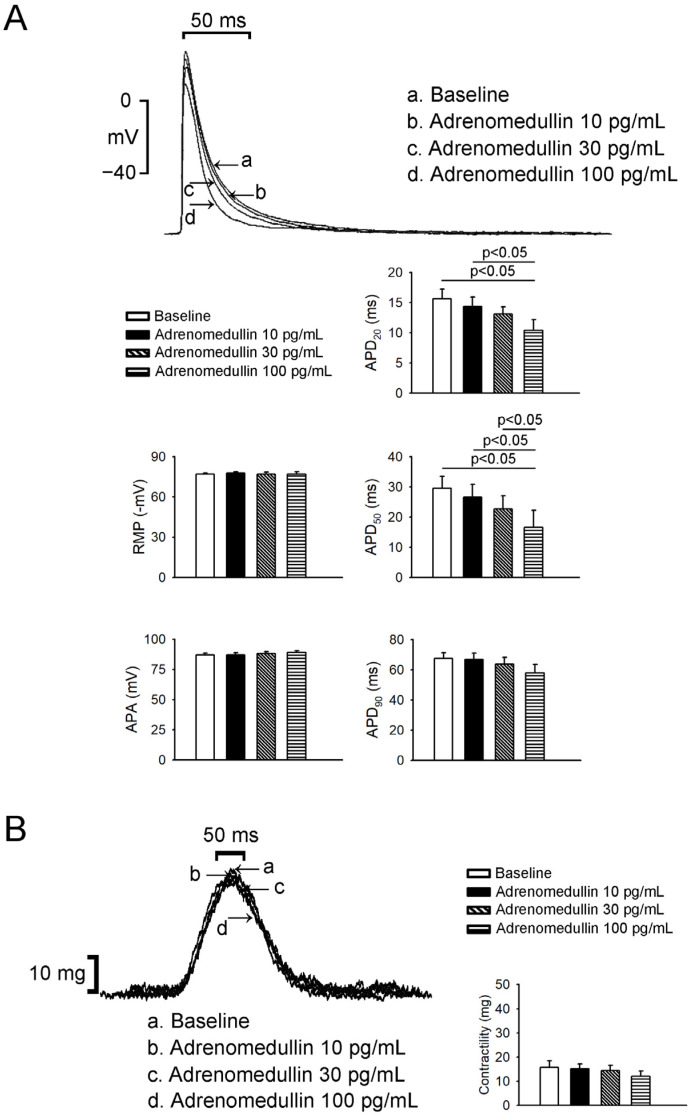
Effects of adrenomedullin on right atrium (RA) AP morphology and contractility. (**A**) The superimposed traces illustrate the effects of adrenomedullin on RA APs. Adrenomedullin (10–100 pg/mL) did not affect APA, MDP, or APD90 in RA tissue (*n* = 6). Adrenomedullin at 100 pg/mL reduced RA AP duration at 20% and 50% repolarization. (**B**) Adrenomedullin did not reduce RA contractility.

**Figure 6 ijms-23-14064-f006:**
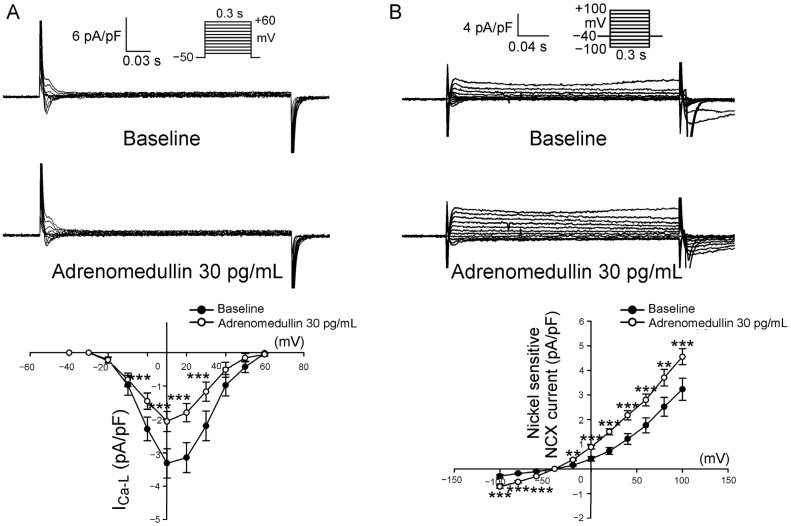
Effects of adrenomedullin on L-type calcium channel (I_Ca-L_) and the sodium–calcium exchanger (NCX) currents in PV cardiomyocytes. (**A**) Current traces and I–V relationship before and after infusion of PV cardiomyocytes (*n* = 13) with 30 pg/mL adrenomedullin. (**B**) NCX current traces and I–V relationship before and after infusion of PV cardiomyocytes (*n* = 12) with 30 pg/mL adrenomedullin. ** *p* < 0.01, *** *p* < 0.005.

## Data Availability

Any data or material that support the findings of this study can be made available by the corresponding author upon request.

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
