# Peer review of "Effects of Adrenomedullin on Atrial Electrophysiology and Pulmonary Vein Arrhythmogenesis"

_ijms, 2022, doi:10.3390/ijms232214064_

Round 1

Reviewer 1 Report

Adrenomedullin is a peptide which has received little notice from the pharmaceutical world since its discovery in the 1990s. It is not clear why there has been little development.   The authors have carried out a credible study assessing the arrythmogenic suppression of adrenomedullin in the pulmonary vein, now thought to be a major source of transition to atrial fibrillation.  The description of methods is excellent.

Although not pertinent to their main theme, they should mention the other known effects of adrenomedullin, including gastrointestinal motility and anti-microbial properties.

The manuscript would be improved by inclusion of several more background articles including:

Martínez-Herrero S, Martínez A. Adrenomedullin: Not Just Another Gastrointestinal Peptide. Biomolecules. 2022 Jan 18;12(2):156.

Voors et al. Adrenomedullin in heart failure: pathophysiology and therapeutic application. Eur J Heart Fail. 2019 ;21:163-171

Ishimitsu et al. Plasma levels of adrenomedullin, a newly identified hypotensive peptide, in patients with hypertension and renal failure. J Clin Invest. 1994 ;94:2158-61

Kohno et al. Plasma Adrenomedullin Concentrations in Essential Hypertension. Hypertension 1996: 27:102-107

The most serious criticism is failure to document adrenomedullin concentration in normal and diseased states.  That must be done to interpret the significance of a failure of 10 pg/ml to affect arrythmogenicity in their studies, although 30 pg/ml had a moderate effect.

Author Response

Thank you very much for your detailed comments. These comments were very instructive and very helpful to this manuscript and to our future research. The responses to these comments are enumerated below:

  1. Regarding the comment “Although not pertinent to their main theme, they should mention the other known effects of adrenomedullin, including gastrointestinal motility and anti-microbial properties. The manuscript would be improved by inclusion of several more background articles including:
  2. Martínez-Herrero S, Martínez A. Adrenomedullin: Not Just Another Gastrointestinal Peptide. Biomolecules. 2022 Jan 18;12(2):156.
  3. Voors et al. Adrenomedullin in heart failure: pathophysiology and therapeutic application. Eur J Heart Fail. 2019 ;21:163-171

III.   Ishimitsu et al. Plasma levels of adrenomedullin, a newly identified hypotensive peptide, in patients with hypertension and renal failure. J Clin Invest. 1994 ;94:2158-61

  1. Kohno et al. Plasma Adrenomedullin Concentrations in Essential Hypertension. Hypertension 1996: 27:102-107”

We appreciated this comment very much. According to your suggestions, we have added these informative references as the background in the revised manuscript (page1-2, line 42-45 and References 3-7, in red font) as follows “Adrenomedullin has several cardiovascular effects (vasodilatation, natriuresis, diuresis, promotion of vascular integrity, and prevention of vascular leakage) and non-cardiovascular effects (anti-inflammatory activity, mucosal epithelial repair, and maintenance of intestinal barrier function).”

  1. Regarding the comment “The most serious criticism is failure to document adrenomedullin concentration in normal and diseased states. That must be done to interpret the significance of a failure of 10 pg/ml to affect arrythmogenicity in their studies, although 30 pg/ml had a moderate effect.”

We appreciated this comment very much. Previous studies have shown that the average plasma concentrations of adrenomedullin were around 7.2 pg/mL in healthy individuals (Toshihiro Kita et al., Drug Des Devel Ther.2020) and around 33.8 pg/mL in patients with heart failure (Jozine M. ter Maaten et al., European Journal of Heart Failure. 2019), respectively. The treatment of adrenomedullin may increase the concentrations of serum adrenomedullin to three times higher than that at baseline (Noritoshi Nagaya et al., et al. circulation.2000). Thus, the concentrations of adrenomedullin used in this study were considered clinically relevant. We have discussed these findings in the revised manuscript (page7, line 149-154 in red font).

Reviewer 2 Report

1.     In the section of Conclusion, the writing is too brief, and it is recommended to discuss in depth.

2.     The reference number inserted in the manuscript should be placed before rather than after the punctuation. For example, in line 43, “diuresis.[1, 2]” should be “diuresis [1, 2].”

3.     In line 137, a full stop is missing after “adrenomedullin”, please add it.

4.     In line 197, “adrenomedullin with and without 100 μM” should be “adrenomedullin with or without 100 μM”.

5.     In line 227, “CsOH).” should be “CsOH.”, please remove the extra parenthesis.

6.     In line 238, “A P value of <.05 was considered statistically significant” suggest to “P value <0.05 was considered statistically significant”.

Author Response

Responses to Reviewer 2

Thank you very much for your detailed comments. These comments were very instructive and very helpful to this manuscript and to our future research. The responses to these comments are enumerated below:

  1. Regarding the comment “In the section of Conclusion, the writing is too brief, and it is recommended to discuss in depth.”

We appreciated this comment very much. According to your suggestions, we have revised the manuscript (page9, line 247-250 in red font) as follows “Adrenomedullin may activate NO signaling and exert antiarrhythmic effects by reducing spontaneous PV activity through mechanoelectrical feedback, reducing ICa-L and NCX current. Adrenomedullin administration represents a potential novel strategy for management of atrial arrhythmogenesis.”

  1. Regarding the comment “The reference number inserted in the manuscript should be placed before rather than after the punctuation. For example, in line 43, “diuresis.[1, 2]” should be “diuresis [1, 2].””

We really apologize for the wrong reference number inserted in the manuscript. We have changed the representative reference number inserted in the revised paper.

  1. Regarding the comment “In line 137, a full stop is missing after “adrenomedullin”, please add it.”

Thank you for your comment. We are sorry for this missing. We have corrected it in the revised paper. (page7, line 139 in red font)

  1. Regarding the comment “In line 197, “adrenomedullin with and without 100 μM” should be “adrenomedullin with or without 100 μM”.”

We appreciated this comment very much. According to your suggestions, we have corrected it in the revised manuscript (page 8, line 204 in red color).

  1. Regarding the comment “In line 227, “CsOH).” should be “CsOH.”, please remove the extra parenthesis.”

We are very sorry for this mistake. We have corrected it in the revised paper. (page9, line 234).

  1. Regarding the comment “In line 238, “A P value of <.05 was considered statistically significant” suggest to “P value <0.05 was considered statistically significant”.”

We appreciated this comment very much. We have corrected it in the revised manuscript (page 9, line 245 in red color).